# Predicting need for heart failure advanced therapies using an interpretable tropical geometry-based fuzzy neural network

Yufeng Zhang[1]*, Keith D. Aaronson[2], Jonathan Gryak[3], Emily Wittrup[1], Cristian Minoccheri[1], Jessica R. Golbus[2], Kayvan Najarian[1,4,5,6]

1 Department of Computational Medicine and Bioinformatics, University of Michigan, Ann Arbor, Michigan, United States of America, 2 Department of Internal Medicine, Division of Cardiovascular Medicine, University of Michigan, Ann Arbor, Michigan, United States of America, 3 Department of Computer Science, Queens College, City University of New York, New York, New York, United States of America, 4 Michigan Institute for Data Science, University of Michigan, Ann Arbor, Michigan, United States of America, 5 Department of Electrical Engineering and Computer Science, University of Michigan, Ann Arbor, Michigan, United States of America, 6 Department of Emergency Medicine, University of Michigan, Ann Arbor, Michigan, United States of America

* chloezh@umich.edu

## Abstract

### Background

Timely referral for advanced therapies (i.e., heart transplantation, left ventricular assist device) is critical for ensuring optimal outcomes for heart failure patients. Using electronic health records, our goal was to use data from a single hospitalization to develop an interpretable clinical decision-making system for predicting the need for advanced therapies at the subsequent hospitalization.

### Methods

Michigan Medicine heart failure patients from 2013–2021 with a left ventricular ejection fraction $\leq$ 35% and at least two heart failure hospitalizations within one year were used to train an interpretable machine learning model constructed using fuzzy logic and tropical geometry. Clinical knowledge was used to initialize the model. The performance and robustness of the model were evaluated with the mean and standard deviation of the area under the receiver operating curve (AUC), the area under the precision-recall curve (AUPRC), and the F1 score of the ensemble. We inferred membership functions from the model for continuous clinical variables, extracted decision rules, and then evaluated their relative importance.

### Results

The model was trained and validated using data from 557 heart failure hospitalizations from 300 patients, of whom 193 received advanced therapies. The mean (standard deviation) of AUC, AUPRC, and F1 scores of the proposed model initialized with clinical knowledge was 0.747 (0.080), 0.642 (0.080), and 0.569 (0.067), respectively, showing superior predictive performance or increased interpretability over other machine learning methods. The model

**Data Availability Statement:** The human research participant data we analyzed in the manuscript contains sensitive information which cannot be shared due to ethic restrictions. However, for

researchers who meet the criteria for access to confidential data, data are available from the University of Michigan's Innovation Partnerships (UMIP) office upon request (contact innovationpartnerships@umich.edu).

**Funding:** Research was supported by the National Science Foundation (Award 2014003). The funders had no role in study design, data collection and analysis, decision to publish, or preparation of the manuscript.

**Competing interests:** Dr. Golbus receives funding from the NIH (L30HL143700) and receives salary support by an American Heart Association grant (grant number 20SFRN35370008). This does not alter our adherence to PLOS ONE policies on sharing data and materials.

learned critical risk factors predicting the need for advanced therapies in the subsequent hospitalization. Furthermore, our model displayed transparent rule sets composed of these critical concepts to justify the prediction.

## Conclusion

These results demonstrate the ability to successfully predict the need for advanced heart failure therapies by generating transparent and accessible clinical rules although further research is needed to prospectively validate the risk factors identified by the model.

## Introduction

Heart failure (HF) affected 6.5 million adults in the United States in 2017 and is expected to impact 8 million by 2030 [1,2]. While medical and device therapies have improved outcomes over the past four decades, 5-year mortality for heart failure remains high at nearly 50% [1,3]. Amongst patients with HF, approximately 5% annually progress to an advanced disease state also known as Stage D HF [4]. For these patients, heart transplantation (HT) and left ventricular assist devices (LVADs) [5–8] offer the best opportunities for long-term survival with improved quality of life. HF advanced therapies, however, carry risks for adverse events and there exists a supply-demand mismatch for donor hearts, necessitating careful patient selection [9]. Therefore, whether and when to refer a HF patient for advanced therapies relies on timely clinical judgment requiring an effective and efficient method of identifying potential candidates amongst all HF patients.

With the high prevalence of HF in the United States, most patients are cared for by general cardiologists or primary care clinicians who may lack specialized training in HF advanced therapies [10]. However, the capacity of advanced HF specialists to evaluate patients for advanced therapies is finite. There is thus a need for decision support systems that can identify patients with HF in need of advanced therapies to ensure that they are referred to a HF cardiologist at the appropriate time. While several risk models have been developed to assist in risk stratification of patients with advanced HF, their focus has been on predicting HF hospitalizations and mortality, typically at pre-specified time points such as 1 or 5 years, rather than assisting providers at the bedside with the timing of advanced therapy delivery. In addition, a subset incorporates data types not collected in routine practice [11–13]. Machine learning has gained significant popularity in recent years and has found applications across various domains, such as biology, medicine, and healthcare, as evident from numerous studies [14–18]. Notably, recent advancements in machine learning have led to successful models for identifying high-risk patients [19–21]. However, these models come with certain limitations, primarily the challenge of interpretability and transparency in model recommendations. This lack of interpretability, often referred to as the "black box" aspect of traditional machine learning models, may pose challenges in gaining acceptance from healthcare providers and in their subsequent integration into healthcare practices. Although some machine learning methods provide a way to interpret the importance of every feature (interpretability), most of the models are not transparent to present the decision logic in a rule format (transparency). Consequently, there is an urgent need for more transparent risk prediction models that can be broadly implemented within electronic health records (EHRs) that use routinely collected data to predict need for HF advanced therapies. This can be used to ensure that HF patients are

referred to advanced HF cardiologists at the appropriate time or to prompt HF cardiologists to initiate a timely advanced therapies evaluation.

Recently, an interpretable algorithm based on a tropical geometry-based fuzzy neural network (TGFNN) was developed [22]. Unlike traditional machine learning methods, this model incorporates existing clinical knowledge and produces a set of criteria by which to explain the rationale for its recommendations [22,23]. We extend that early work herein from classification to risk prediction, predicting the future need for HF advanced therapies using routinely collected clinical variables from a single hospitalization.

## Methods

The study utilized a TGFNN to identify patients who would require advanced therapies for heart failure during a subsequent hospitalization. The proposed system's flow diagram is depicted in Fig 1A. Prior to analysis, the study obtained approval from the University of Michigan Institutional Review Board (HUM00184418) which waived the need for informed consent, and the EHR data used in the study were completely deidentified prior to analysis. The data was accessed from Michigan Medicine on May 18th, 2021.

### Study cohort

Eligible patients were 18–80 years of age at the time of a hospitalization at Michigan Medicine for acute on chronic HF, as derived from billing codes, where they received at least one dose of an intravenous diuretic. Qualifying hospitalizations were between January 1, 2013, and January 30, 2021. Patients were included if their most recent ejection fraction recorded in the EHR at the time of admission was less than or equal to 35% and excluded if they had a body mass index (BMI) > 50 kg/m$^2$. Patients were required to have at least two eligible HF hospitalizations in one year to support the study design.

To predict the next-visit state, eligible pairs of HF hospitalizations were then labeled as positive if the patient received an urgent HT or LVAD during their second hospitalization, with urgent HTs defined as those transplanted at Status 1A or 1B prior to October 17, 2018, or at statuses 1–4 thereafter. The remaining hospitalization pairs were labeled as negative and included those too well for HF advanced therapies, which we defined as patients who survived at least two years after their first HF hospitalization without the need for HT or LVAD implantation. The resulting cohort consisted of 557 HF hospitalizations pairs (samples) from 300 patients, 193 of whom received advanced therapy at their second hospitalization. Within our study cohort, 157 patients had two encounters, 66 three encounters, and the remaining patients more than three encounters. The time intervals between the two consecutive visits exhibit the following distribution: the ¼ quantile is 27 days, the ½ quantile is 61 days, and the ¾ quantile is 189 days.

### Clinical features and preprocessing

Our model incorporated continuous and categorical clinical variables from each hospitalization identified by HF cardiologists as being of clinical value in the setting of HF, including laboratory values, vital signs, and comorbidities as determined by Elixhauser [24] (S1 and S2 Tables in S1 File). For most continuous variables, we only utilized the first measurement obtained during a given hospitalization. For brain natriuretic peptide (BNP) and creatinine, the relative change over each hospitalization (first to last measurement) was expressed as the percent change over the hospitalization. Furthermore, the first measurement of both systolic and diastolic blood pressure (BP) was used to calculate mean arterial pressure (MAP) and pulse pressure. Standardization was executed for all continuous features after data partition,

A

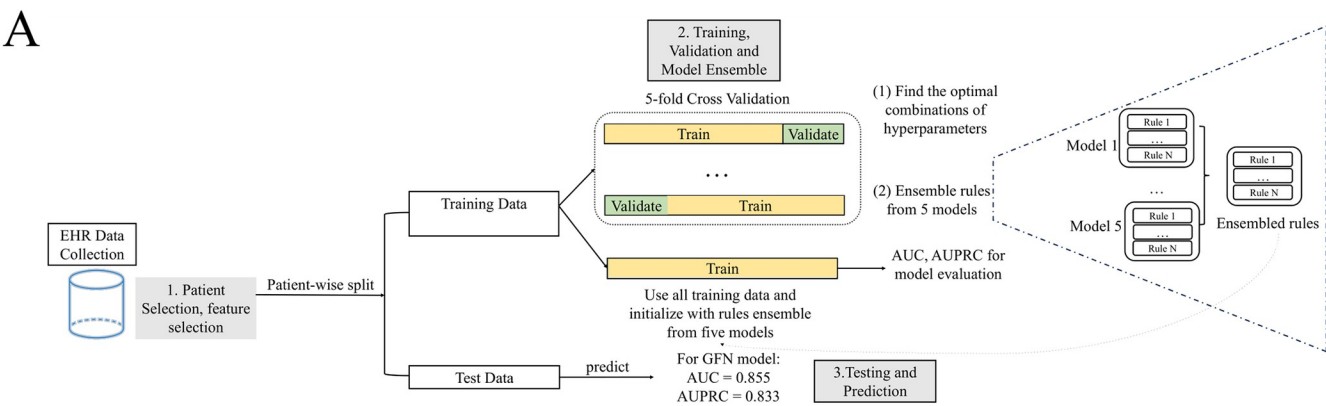

B

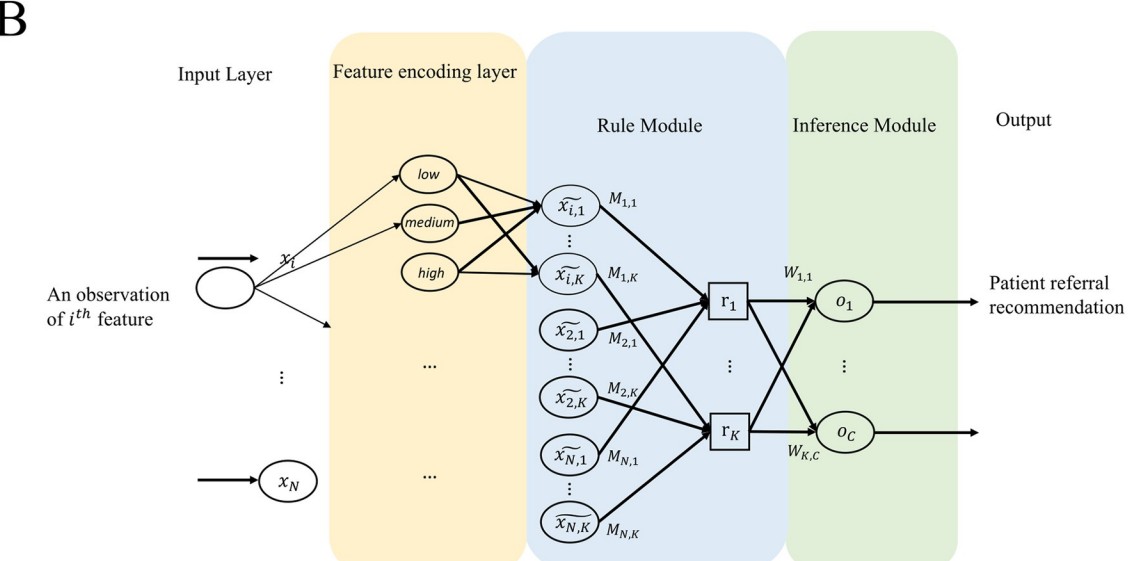

**Fig 1. Flowchart of the clinical decision support system and the interpretable tropical geometry-based fuzzy neural network algorithm.** Fig 1A describes the system and training strategy: The EHR dataset is collected from Michigan Medicine and then the patient selection and outcome definition were performed. The data is split into training and test sets. Five-fold cross-validation is performed on the training dataset. Rules extracted from five-folds are ensembled. The model is then retrained on the whole training dataset with ensembled rule initialization. The trained model is later validated on the test set. Fig 1B depicts the structure of tropical geometry-based fuzzy neural network: The encoding layer encodes the input features into 'low', 'medium' and 'high' fuzzy concepts. The rule layer combines different concepts to generate several rules and decisions are made at the final inference layer by leveraging all rules. The edges between modules are trainable parameters to optimize the model. $x_i$: continuous variables; $l_{\epsilon_1}(x_i)$: low concept membership function; $m_{\epsilon_1}(x_i)$: medium concept function; $h_{\epsilon_1}(x_i)$: high concept membership function; A: attention matrix; M: connection matrix; W: inference matrix.

while one-hot encoding was employed for all categorical features to transform them into binary or ordinal representations. Data imbalance was not addressed during this phase; instead, weighted loss was implemented throughout the entire training process by assigning higher weights to positive samples.

For missing values, carry-forward imputation was applied between encounters, and any remaining missing values were imputed using multiple imputations [25]. Although echocardiographic features had an approximately 40% missing rate, they were retained due to their importance in HF decision-making. Any other features with a missing rate greater than 60% were removed, and any patient with more than ten missing values was excluded from the analysis.

## Tropical geometry-based interpretable machine learning method

We used the TGFNN to predict the future need of advanced therapies. While the TGFNN model has demonstrated successful application in the classification task using the REVIVAL and INTERMACS registries, its potential in the prediction task with EHR data remains unexplored. In this proof-of-concept study for the prediction task, we utilize EHR data from previous hospitalizations as input. The overall architecture of the algorithm is depicted in Fig 1B. The TGFNN algorithm consists of three modules: encoding, rule extraction, and inference. In the encoding module, each comorbidity is assigned a value of 1 if it exists in the patient's medical history and 0 otherwise with an indicator function. At the same time, continuous variables are encoded into three fuzzy concepts: 'low', 'medium' and 'high'. Fuzzy concepts offer a valuable approach to address the complexities and uncertainties associated with determining cutoff points in clinical practice by encoding continuous variables into three categories, thereby accommodating the inherent ambiguity in defining thresholds. The three fuzzy concepts correspond to membership functions $l_{\varepsilon_1}(x_i)$, $m_{\varepsilon_1}(x_i)$ and $h_{\varepsilon_1}(x_i)$, which are defined as follows:

$$f_{\varepsilon_1}(x_i) = \varepsilon_1 \, log(1 + exp(x_i/\varepsilon_1)) \tag{1}$$

$$l_{\varepsilon_1}(x_i) = f_{\varepsilon_1}\left(\frac{a_{i,2} - x_i}{a_{i,2} - a_{i,1}}\right) - f_{\varepsilon_1}\left(\frac{a_{i,1} - x_i}{a_{i,2} - a_{i,1}}\right) \tag{2}$$

$$m_{\varepsilon_1}(x_i) = f_{\varepsilon_1}\left(\frac{x_i - a_{i,1}}{a_{i,2} - a_{i,1}}\right) - f_{\varepsilon_1}\left(\frac{x_i - a_{i,2}}{a_{i,2} - a_{i,1}}\right) - f_{\varepsilon_1}\left(\frac{a_{i,3} - x_i}{a_{i,4} - a_{i,3}}\right) + f_{\varepsilon_1}\left(\frac{a_{i,4} - x_i}{a_{i,4} - a_{i,3}}\right) - 1 \tag{3}$$

$$h_{\varepsilon_1}(x_i) = f_{\varepsilon_1}\left(\frac{x_i - a_{i,3}}{a_{i,4} - a_{i,3}}\right) - f_{\varepsilon_1}\left(\frac{x_i - a_{i,4}}{a_{i,4} - a_{i,3}}\right) \tag{4}$$

where $a_{i,j}$ denotes the cutoff parameters for the concepts and $\varepsilon$ controls the smoothness of membership functions. The membership function defines to what degree the measurements belong to each concept rather than trichotomizing variables that exist along a continuum. Therefore, the uncertainty that fuzzy concepts introduce can make the model more flexible in terms of interpretability. In addition, the 'cut-off points' for three fuzzy concepts are learned from the study cohort by the algorithm rather than pre-defined.

The rule module consists of two layers. The first layer leverages the three concepts of variable $i$ in relation to rule $k$ using attention tensor $\mathbf{A} \in \mathbb{R}^{N \times 3 \times K}$. Each entry $A_{i,j,k}$ shows the importance of $x_i$ being of concept $j$ to the rule $k$. The message passing formula for this layer is given as

$\tilde{x}_{i,k} = A_{i,1,k}l(x_i) + A_{i,2,k}m(x_i) + A_{i,3,k}h(x_i)$. Every entry is normalized to [0, 1] and trainable. High value in $\mathbf{A}$ indicates the higher importance in the decision system. The second layer measures the importance of $i$-th variable *to $k$-th* rule using a connection matrix $\mathbf{M}$ of size $N \times K$, whose entries are also normalized between [0,1] and trainable. The second half of the message-passing formula of the rule module is given by $r_k = T_\varepsilon(\tilde{x}_{1,k}^{M_{1,k}}, \tilde{x}_{2,k}^{M_{2,k}} \ldots \tilde{x}_{N,k}^{M_{N,k}})$. $r_k$ measures the rule firing strength using a parameterized T-norm. The parameterized T-norm with two

inputs is define by

$$T_{\varepsilon_2}(x,y) = g_{\varepsilon_2}^{-1}(g_{\varepsilon_2}(x) + g_{\varepsilon_2}(y)) = \left(x^{\frac{\varepsilon_2-1}{\varepsilon_2}} + y^{\frac{\varepsilon_2-1}{\varepsilon_2}} - 1\right)^{\frac{\varepsilon_2}{\varepsilon_2-1}} \tag{5}$$

$$g_{\varepsilon_2}(y) = \frac{\varepsilon_2}{1-\varepsilon_2}\left(1 - x^{\frac{\varepsilon_2-1}{\varepsilon_2}}\right) \tag{6}$$

When $\varepsilon_2$ approaches to 1, parameterized T-norm approaches to the multiplication operation while $\varepsilon_2$ approaches to 0, parameterized T-norm takes the minimum of the inputs. The higher value in M, the higher contribution to the firing strength. The N-input T-norm is defined as:

$$r_k = \left(\sum_{i=1}^{H} \tilde{x}_{i,k}^{M_{i,k}\frac{\varepsilon_2-1}{\varepsilon_2}} - N + 1\right)^{\frac{\varepsilon_2}{\varepsilon_2-1}} \tag{7}$$

In the inference module, an inference matrix $W$ of size $K{\times}C$ is learned to estimate the rule firing strengths, where C denotes the number of outcome classes. Each entry $W_{k,c}$ of the inference matrix represents the contribution of the rule $k$ to the final prediction of the class c, which is calculated as $O_c = Q_{\varepsilon_3}(W_{1,c}r_1, W_{2,c}r_2 \ldots, W_{K,c}r_K)$. The parametrized T-conorm $Q_{\epsilon_3}$ with two inputs can be defined as $Q_{\varepsilon_3}(x,y) = \left(x^{\frac{1}{\varepsilon_3}} + y^{\frac{1}{\varepsilon_3}}\right)^{\varepsilon_3}$. When $\varepsilon_3 \rightarrow 1$, parameterized T-conorm approaches to the addition operation while $\varepsilon_3 \rightarrow 0$, parameterized T-conorm takes the maximum of the inputs. The two-inputs T-conorm $Q_{\varepsilon_3}$ can be generalized to K-input T-conorm

$$Q_{\varepsilon_3} : O_c = \left(\sum_{k=1}^{K} (W_{k,c}r_k)^{\frac{1}{\varepsilon_3}}\right)^{\varepsilon_3} \tag{8}$$

The three smoothness parameters $\varepsilon_1, \varepsilon_2, \varepsilon_3$ are all trainable with constraint $0 < \varepsilon_1, \varepsilon_2, \varepsilon_3 < 1$ and can control the (1) sharpness of the membership function (2) the behavior of the T-norm and T-conorm functions. In our current study, we used the same $\varepsilon$ for simplicity. All parameters were trained using Adam optimizer except for $\varepsilon$. $\varepsilon$ is initialized with 0.99 and then decrease at every training step using the scheduling formula: $\varepsilon = \max(\varepsilon_{min}, \varepsilon \cdot \gamma^{training\_step})$ where $\gamma$ is the decay rate. Furthermore, one-hot encoded categorical features do not require membership functions, but their category levels behave like the three concepts of continuous variables. Therefore, the weighting process in the rule and inference modules can be applied as well to the categorical variables if we adapt the number of concepts from 3 to the number of category levels.

The algorithm was trained with a weighted cross entropy alongside two regularization terms through backpropagation. The first regularization term is defined as below in favor of feature sparsity:

$$loss_{sparse} = \| vec(A) \|_1 + \| vec(M) \|_1 \tag{9}$$

The second regularization term add penalty for highly correlated rules and is formulated as:

$$loss_{corr} = \sum_{i=1}^{N-1} \sum_{j=i+1}^{N} vec(S_{:,:,i})vec(S_{:,:,j}) \tag{10}$$

where **S** is constructed utilizing the entries in the attention matrix, **A** and connection matrix **M** as follows: $S_{i,d,k} = A_{i,d,k}{\times}M_{i,k}$, where $i \in \{1,\ldots N\}$, $d \in \{1,2,3\}$, $k \in \{1,\ldots K\}$. The contribution matrix **S** represents the contributions of individual concepts and variables to each rule. The

total loss function can be therefore written as:

$$loss = loss_{weighted\ cross\_entropy} + \lambda_1 loss_{sparse} + \lambda_2 loss_{corr} \tag{11}$$

Where $vec(\cdot)$ denotes matrix vectorization.

Through the attention matrix and connection matrix, the algorithm operates with complete transparency, enabling the extraction and display of the underlying rules, thus revealing the overall decision-making logic.

## TGFNN rule initialization, ensemble, and extraction

To enhance the TGFNN model with clinical knowledge, we gathered four simplified rules from HF cardiologists and from the medical literature [1,2,26]. Rules used for network initialization are provided below:

- **IF** left ventricular ejection fraction (LVEF) is *low* **AND** systolic BP (SBP) is *low*, **THEN** refer to HT/ LVAD

- **IF** LVEF is *low* **AND** mitral regurgitation is *high (severe)* **THEN** refer to HT/ LVAD

- **IF** LVEF is *low* **AND** BNP change *elevated* (positive delta from first to last measurements during the initial hospitalization) **THEN** refer to HT/ LVAD

- **IF** LVEF is *low* **and** serum sodium is *low* **THEN** refer to HT/ LVAD

These rules were used to initialize the network for every fold prior to training. The resulting learned rules were filtered based on their firing strength and correlations with one another, while the corresponding features were selected by their contribution to each individual rule. Thus, the highest-weighted and least-correlated rules with important variables were retained and ensembled for network re-initialization. Details are illustrated in the supporting information.

## Experimental design

This study compared the TGFNN to the following classical machine learning models: Random Forest, XGBoost, Logistic Regression, Support Vector Machines, Naive Bayes, and Decision Trees on the same dataset. We performed patient-wise five-fold cross-validation on the training dataset to evaluate model performance and robustness. A random search algorithm for hyperparameter tuning was employed during the training stage. The models were then evaluated on the validation and test datasets, which were kept separate from the training dataset. In our experimental setting, the validation sets were utilized to assess the model's performance and robustness as an unseen dataset with the same data distribution as the training dataset. The test set served as an external dataset for overall model evaluation. The details of the data split are further described in supplementary materials.

## Model evaluation

Models were then evaluated by using the mean (standard deviation [SD]) and by their accuracy ((true positive + true negative) / (all positives + all negatives)), recall (true positives/(true positives + false negatives)), specificity, precision (true positives/(true positives + false positives)), F1 score (harmonic mean of precision and recall), area under the receiver operating curve (AUC), area under the precision-recall curve (AUPRC) and Matthews correlation coefficient (MCC) [27]. Each of the machine learning models was also assessed concerning (1)

**Table 1. Demographic characteristics of patients requiring HT/LVAD evaluation ("Positive") and those too well for HF advanced therapies ("Negative").** Displayed are mean (standard deviation) for continuous variables or N (%) for categorical variables.

| | Total (N = 557) | "Positive" (N = 193) | "Negative" (N = 364) |
|---|---|---|---|
| **Male Gender** | 413 (74.1%) | 157 (81.3%) | 256 (70.3%) |
| **Age, years** | 56±14 | 53 ± 14 | 58 ± 14 |
| **SBP, mmHg** | 115.11 ± 22.28 | 102.74 ± 14.95 | 121.68 ± 22.74 |
| **MAP, mmHg** | 85.07±15.97 | 76.97 ± 11.89 | 89.36 ± 16.22 |
| **Serum Sodium, mmol/L** | 137.86±4.10 | 137.48 ±3.58 | 138.06 ± 4.35 |
| **Serum Potassium, mmol/L** | 4.24±0.54 | 4.25 ± 0.57 | 4.23 ± 0.52 |
| **Delta creatinine, %** | 1.20 ± 18.75 | 1.20 ± 19.25 | 1.20 ± 18.54 |
| **Delta BNP, %** | 15.70 ± 91.53 | 12.34 ± 57.56 | 17.51 ± 103.26 |
| **LVEF, %** | 19.85±8.38 | 17.59 ± 9.67 | 21.06 ± 8.39 |
| **LVIDd2D, mm** | 67.40±11.28 | 69.46 ± 11.79 | 66.06 ± 11.13 |
| **Renal Failure** | 64.60% | 60.10% | 67.00% |
| **Liver Disease** | 31.40% | 28.00% | 33.20% |
| **Chronic Pulmonary Disease** | 79.20% | 69.40% | 84.30% |

Abbreviations: SBP: Systolic blood pressure; BNP: Brain natriuretic peptide.

whether the model could evaluate variable importance (interpretability) and (2) whether model prediction can be explained by clinical rules (transparency).

## Results

### Prediction performance

Patient characteristics are shown in Table 1.

The cross-validation results for both the TGFNN and other standard machine-learning models are summarized in Table 2. The model's performance when initialized with clinical knowledge achieved an F1 score of 0.569, an AUC of 0.747, and an AUPRC of 0.642. Our TGFNN with clinical initialization outperformed all standard machine learning models with respect to their F1 scores, AUC, and AUPRC except for XGBoost and Random Forest, which are ensemble models that lack explicit rules and operate through complex combinations of decision boundaries, making it challenging to interpret their inner workings. These results highlight the advantage of our model regarding transparency, interpretability, and performance in comparison to traditional ensemble approaches.

**Table 2. Performance of machine learning models on HF dataset using 5-fold cross validation.** Models are referred to as transparent if they can explain their recommendations in a way understood by humans. The column 'Interpretability' indicates whether the feature importance can be provided with the model. Although Random Forest, XGBoost and SVM are listed as interpretable, these models can only be interpreted using external approach such as SHAP (SHapley Additive exPlanations). The column "Rules" refers to whether the model provides a set of clinical rules by which to explain its prediction.

| Model | Accuracy | Recall | F1 | AUC | AUPRC | MCC | Interpretability | Rules |
|---|---|---|---|---|---|---|---|---|
| **TGFNN (with domain knowledge)** | 0.744 (0.040) | 0.614 (0.091) | 0.569 (0.067) | 0.747 (0.080) | 0.642 (0.080) | 0.425 (0.088) | Yes | Yes |
| **Logistic Regression** | 0.723 (0.024) | 0.561 (0.059) | 0.530 (0.029) | 0.730 (0.031) | 0.591 (0.077) | 0.379 (0.044) | Yes | No |
| **Naive Bayes** | 0.661 (0.046) | **0.653 (0.12)** | 0.516 (0.058) | 0.724 (0.083) | 0.598 (0.085) | 0.312 (0.083) | Yes | No |
| **Decision Tree** | 0.688 (0.021) | 0.574 (0.12) | 0.501 (0.083) | 0.650 (0.037) | 0.451 (0.067) | 0.299 (0.066) | Yes | Yes |
| **Random Forest** | **0.772 (0.020)** | 0.505 (0.067) | 0.551 (0.056) | **0.818 (0.031)** | **0.732 (0.018)** | 0.462 (0.053) | Yes | No |
| **XGBoost** | 0.754 (0.052) | 0.606 (0.081) | **0.576 (0.091)** | 0.792 (0.038) | 0.689 (0.065) | 0.442 (0.128) | Yes | No |
| **SVM** | 0.741 (0.032) | 0.431 (0.11) | 0.478 (0.098) | 0.759 (0.051) | 0.630 (0.075) | 0.374 (0.099) | Yes | No |

Table 3. Performance of machine learning models on the HF test dataset.

| Model | Accuracy | Recall | Specificity | Precision | F1 | AUC | AUPRC | MCC |
|---|---|---|---|---|---|---|---|---|
| TGFNN with ensemble initialization | **0.820** | 0.568 | 0.968 | 0.913 | 0.656 | **0.855** | **0.833** | **0.593** |
| Logistic Regression | 0.630 | 0.000 | **1.000** | 0.000 | 0.000 | 0.509 | 0.410 | 0 |
| Naive Bayes | 0.770 | **0.757** | 0.778 | 0.667 | **0.662** | 0.837 | 0.722 | 0.523 |
| Decision Tree | 0.720 | **0.757** | 0.698 | 0.596 | 0.621 | 0.783 | 0.673 | 0.415 |
| Random Forest | 0.790 | 0.568 | 0.921 | 0.808 | 0.621 | 0.842 | 0.798 | 0.537 |
| XGBoost | 0.770 | 0.595 | 0.873 | 0.733 | 0.611 | 0.830 | 0.773 | 0.493 |
| SVM | 0.630 | 0.000 | 1.000 | 0.000 | 0.000 | 0.446 | 0.356 | 0 |

We also assessed model performance on the holdout test dataset as shown in Table 3. The re-initialized TGFNN model with ensembled rules extracted from 5 folds improved the F1 score from 0.577 to 0.656. In addition, the re-initialized TGFNN model achieved the highest AUC (0.855) and AUPRC (0.833) of all machine learning models. In clinical practice, there is a preference for models that can provide their underlying logic, as it enables a better understanding of the reasoning behind their predictions for healthcare professionals.

## TGFNN rules

The rules extracted from the re-initialized TGFNN model are presented in Fig 2, showing how the network makes decisions. Among the seven rules, rules 2–5 were learned from the data apart from the initially injected rules. For example, patients with low systolic blood pressure, low BMI, and low LVEF were more likely to be recommended for heart transplantation.

Based on the learned rules, low SBP, low MAP, and low LVEF on hospital admission are the most important indicators of needing HF advanced therapies. Other factors, such as a relative increase in creatinine or hyponatremia, were also important indicators of need for HF advanced therapies. Demonstrative rules for identifying patients in need of HF advanced therapy are presented below:

- Rule: **IF** MAP is *low* **AND** Creatinine *increases* **AND** LVEF is *low*, **THEN** refer for HT/ LVAD

- Rule: **IF** SBP is *low* **AND** Hemoglobin is *low* **AND** LVEF is *low* **AND** *No* Diabetes THEN refer for HT/ LVAD

## Range inference

Membership functions for several important continuous medical variables drawn from the model are shown in Fig 3, depicting how concepts (low, medium, high) are assigned to clinical variables to generate fuzzy sets in the rules. Furthermore, our model allows the transition between the 'low' and 'medium' concepts for SBP to be smooth and the change between the 'medium' and 'high' concepts to be sharp. Asymmetrical smoothness is vital as it provides greater flexibility and uncertainty in decision-making, which is useful for interpretation, suggesting the different disease progressions at the boundary between the adjacent fuzzy concepts.

We can infer the possible ranges leading to decision-making by utilizing these membership function boundaries. The learned boundaries for these three concepts for clinical features are shown in Table 4. We observe consistency by comparing the learned boundaries and possible reference ranges that HF cardiologists employ in clinical practice.

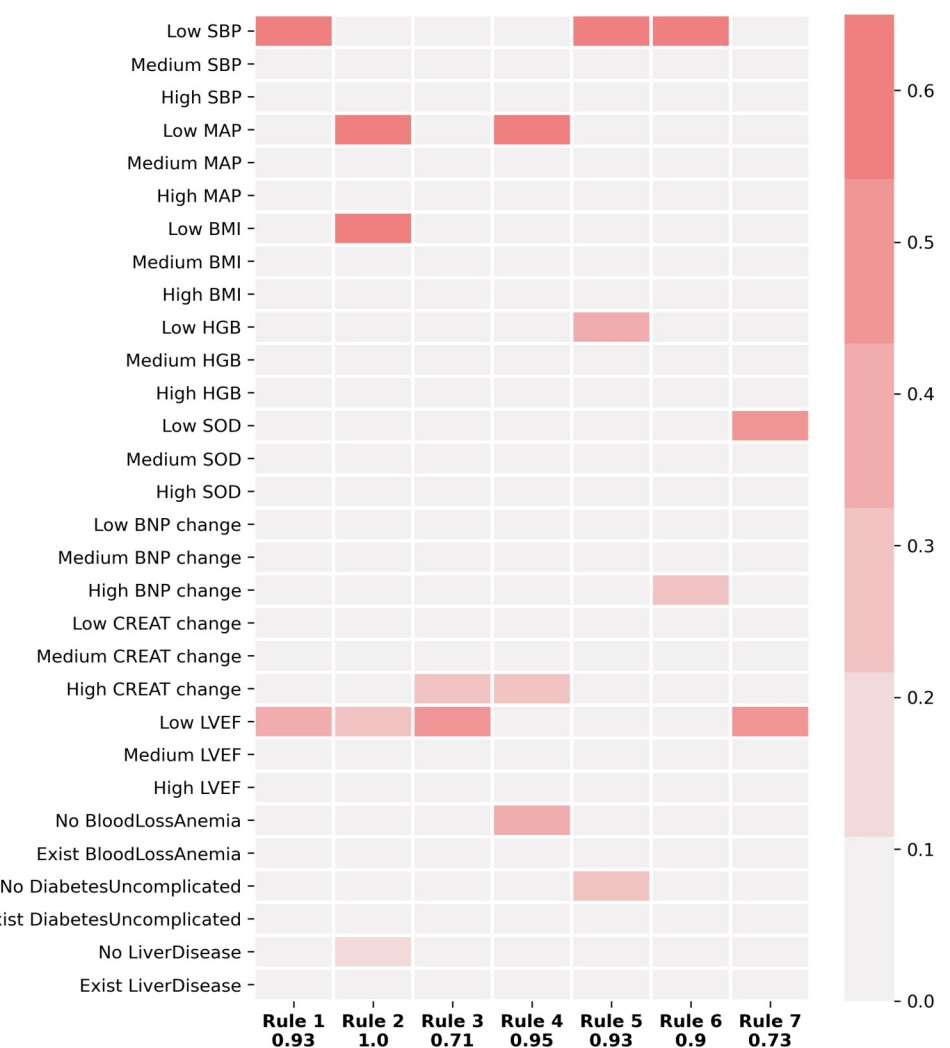

**Fig 2. Clinical rules extracted from the network: In the heatmap, each column represents a rule, while each row represents one concept of a clinical feature.** The number beneath every rule measures the contribution of the rule. The color shades on the heatmap indicate the importance of individual concepts for each rule. Rule 1 can be written as: IF Systolic Blood Pressure is low AND Left Ventricular Ejection Fraction is low, THEN refer for heart transplantation/ LVAD. KEY: BMI = body mass index; BNP = brain natriuretic peptide; CREAT = creatine; HGB = hemoglobin; LVEF = left ventricular ejection fraction; MAP = mean arterial pressure; SBP = systolic blood pressure; SOD = sodium.

## Individual performance

In addition to the population-level rule extraction and predictive performance, rules can also be applied at the individual level. Upon feeding the patient's EHR profile into the model, it not only generates predictions but also highlights the specific rules that are triggered for the given case. We illustrate this by using one patient who was referred for advanced therapies in our test set (Fig 4). The TGFNN with ensemble initialization successfully predicted the patient's need for HF advanced therapies at his subsequent hospitalization. Rules 1, 2, 4, 5 and 6 in Fig 3 activated for this patient, leading to the recommendation for advanced therapies.

## Discussion

Herein we describe a transparent and interpretable machine learning model capable of using EHR data to predict whether HF patients will require advanced therapies at a future HF

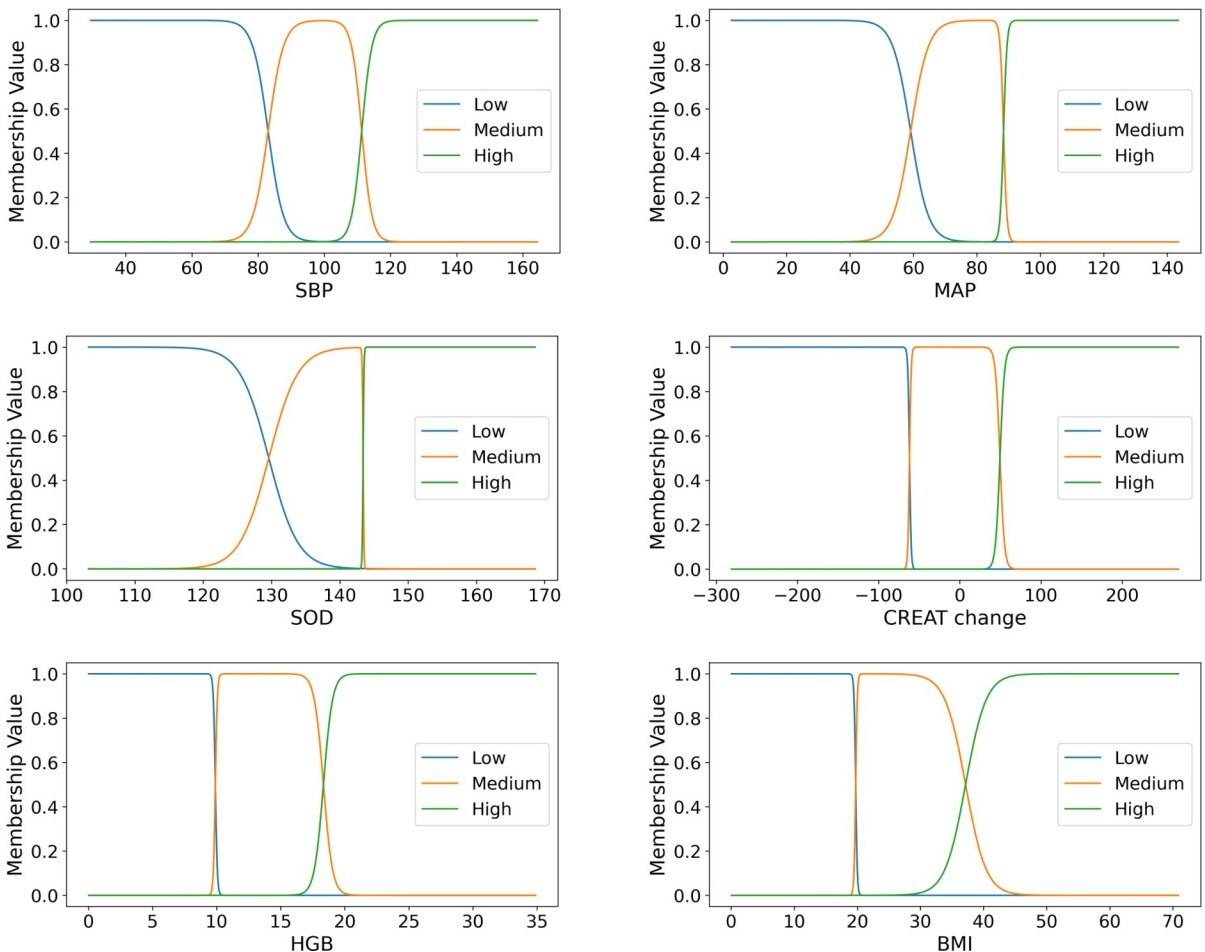

**Fig 3. Membership function visualization: Continuous clinical features are encoded into three concepts: "low', 'medium' and 'high'.** Membership values range from 0 to 1. The x-axis of each membership function represents the range of possible values, while the y-axis represents the degree of membership of each value in the corresponding fuzzy set, ranging from 0 to 1. The X-coordinates of the intersection of two membership functions indicate where the transition from one concept to another occurs KEY: SBP = systolic blood pressure; MAP = mean arterial pressure; SOD = sodium; HGB = hemoglobin; BMI = body mass index; CREAT = creatine.

hospitalization. Our model outperformed all but two of the standard machine learning models to which it was compared and was the only model to be both transparent, providing the rationale for its recommendations, and interpretable. Importantly, unlike prior machine learning

**Table 4. Critical values of membership functions learned from the study cohort by the algorithm.** These critical values indicate the potential threshold where adjacent concepts transition.

| Features | Inference range | | | | |
| --- | --- | --- | --- | --- | --- |
| | low | medium left | medium right | high | unit |
| SBP | 81.8 | 84.3 | 110.5 | 112.1 | mmHg |
| MAP | 57.6 | 60.6 | 88.1 | 88.7 | mmHg |
| Sodium | 128.4 | 130.8 | 143.3 | 143.4 | mmol/L |
| Creatine change | -62.5 | -61.5 | 48 | 50.9 | % |
| HGB | 9.8 | 9.9 | 18.1 | 18.5 | g/dL |
| BMI | 19.7 | 19.8 | 36.3 | 38 | kg/m2 |

**Patient A Profile:**
- SBP: 84 mmHg
- MAP : 70.7 mmHg
- BMI: 22 kg/$m^2$
- SOD:133.5 mmol/L
- HGB: 11.9 g/dL
- ΔBNP :286.6 %
- Δ CREAT:13.2%
- LVEF:15 %
- Anemia: No
- Liver disease: Yes
- Diabetes: No

**Fig 4. Profile for a patient in the test dataset, showing the composite rules that fired.**

models, the current model uses routinely collected EHR data from a single hospitalization to predict the need for HF advanced therapies during a subsequent hospitalization. Such an approach allows the mobilization of critical resources to ensure that patients are able to undergo a comprehensive, advanced therapies evaluation in an anticipatory rather than a reactive manner, with the latter placing patients at risk for clinical deterioration precluding advanced therapies.

In light of the burgeoning numbers of HF patients, there has been growing interest in developing clinical decision-support systems capable of identifying patients with advanced HF [28]. These models have differentiated themselves from many of the historical regression-based models whose limitations have included a focus on mortality and hospitalizations at pre-specified time points, reliance on data not routinely collected in practice, need for a relatively small number of clinical variables, and inability to account for non-linear relationships amongst variables. Recent machine learning models have overcome many of the limitations of these traditional models. These include an augmented intelligence-enabled workflow for identifying outpatients with Stage D HF warranting clinical review to determine need for referral to a HF cardiologist [19] and an ensemble deep learning model trained to predict all-cause death, listing for HT, or extracorporeal membrane oxygenation (ECMO)/VAD within 1-year [20].

Our model distinguished itself from the previous methods in a number of ways. First, the transparent structure of the TGFNN method allowed for the justification of treatment recommendations at both the population and individual levels through fuzzy rules. These rules enable the evaluation of feature importance and feature interaction and can be quickly verified by clinicians and tested for applicability in other clinical settings. Second, the model defined abnormal ranges for continuous variables, aiding in model interpretability and in the utilization of these ranges when caring for patients when clinical decision support may be unavailable. Finally, our model predicted the future need for advanced HF therapies using routinely collected data from a single hospitalization, thereby moving from classification to prediction and avoiding the risk of missing the optimal advanced therapies window.

This study should be interpreted within the context of its limitations. First, the data were obtained from a single medical center, limiting the generalizability of our study findings. Thus, our work will need to be validated in additional settings with a larger sample. Second, the model requires prospective validation using the EHR with subsequent clinician review of model recommendations. Such an approach, when implemented elsewhere, led to an increase

in clinical referrals to HF cardiologists as well as an increase in advanced therapies evaluations [19]. Third, the current algorithm only uses the previous visit to predict whether the patient will subsequently require advanced therapies. Future enhancements of the model will incorporate more extensive longitudinal data, potentially improving model performance. Finally, our analysis only incorporated a subset of the clinical variables with known associations with advanced HF. A greater number of diverse variables will be added to the analysis for future exploration, potentially improving model performance and allowing the generation of additional clinical rules.

In conclusion, in this study, a TGFNN, an interpretable and transparent machine learning method, was applied to predict the future need for HF advanced therapies using data routinely collected in the EHR. The results show that this method's performance exceeds existing traditional machine learning methods while extracting clinical rules that are easily interpretable and verifiable. Future research is needed, however, to incorporate longitudinal data and a broader sample of HF patients for long-term prediction.

## Supporting information

**S1 File. Contains the training details, clinical characteristic of patient encounters from Michigan Medicine and the data split information.**
(DOCX)

## Author Contributions

**Conceptualization:** Yufeng Zhang, Keith D. Aaronson, Jessica R. Golbus, Kayvan Najarian.

**Data curation:** Yufeng Zhang, Jonathan Gryak.

**Formal analysis:** Yufeng Zhang.

**Funding acquisition:** Keith D. Aaronson, Jessica R. Golbus, Kayvan Najarian.

**Investigation:** Keith D. Aaronson, Jessica R. Golbus, Kayvan Najarian.

**Methodology:** Yufeng Zhang.

**Project administration:** Keith D. Aaronson, Jonathan Gryak, Emily Wittrup, Cristian Minoccheri, Kayvan Najarian.

**Resources:** Keith D. Aaronson, Jonathan Gryak, Jessica R. Golbus, Kayvan Najarian.

**Software:** Yufeng Zhang, Jonathan Gryak.

**Supervision:** Keith D. Aaronson, Emily Wittrup, Jessica R. Golbus, Kayvan Najarian.

**Validation:** Yufeng Zhang, Jessica R. Golbus.

**Visualization:** Yufeng Zhang.

**Writing – original draft:** Yufeng Zhang, Keith D. Aaronson, Jessica R. Golbus.

**Writing – review & editing:** Yufeng Zhang, Keith D. Aaronson, Emily Wittrup, Jessica R. Golbus.

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
