## [Decision Letter · Decision Letter 0]

11 Sep 2023

PONE-D-23-19605Predicting Need for Heart Failure Advanced Therapies using an Interpretable Tropical Geometry-based Fuzzy Neural NetworkPLOS ONE

Dear Dr. Zhang,

Thank you for submitting your manuscript to PLOS ONE. After careful consideration, we feel that it has merit but does not fully meet PLOS ONE’s publication criteria as it currently stands. Therefore, we invite you to submit a revised version of the manuscript that addresses the points raised during the review process. Please address the points raised by the reviewer. Additionally, please provide more details about the method used to analyse the data, analysis results and their interpretation.

We look forward to receiving your revised manuscript.

Kind regards,

Suyan Tian

Academic Editor

PLOS ONE

“Research was supported by the National Science Foundation (Award 2014003)”

“Dr. Golbus receives funding from the NIH (L30HL143700) and receives salary support by an American Heart Association grant (grant number 20SFRN35370008).”

Reviewers' comments:

Reviewer's Responses to Questions

**Comments to the Author**

1. Is the manuscript technically sound, and do the data support the conclusions?

Reviewer #1: Yes

2. Has the statistical analysis been performed appropriately and rigorously? 

Reviewer #1: Yes

3. Have the authors made all data underlying the findings in their manuscript fully available?

Reviewer #1: Yes

4. Is the manuscript presented in an intelligible fashion and written in standard English?

Reviewer #1: Yes

5. Review Comments to the Author

Reviewer #1: University of Michigan Health patients from 2013-2021 with heart failure, a left ventricular ejection fraction < 35%, and at least two heart failure hospitalizations were used to train an interpretable machine learning model constructed using fuzzy logic and tropical geometry. Clinical knowledge was used to initialize the model. The performance and robustness

of the model were evaluated with the mean and standard deviation of the area under the receiver operating curve (AUC), the area under the precision-recall curve (AUPRC), and the F1 score. We inferred membership functions from the model for continuous clinical variables, extracted decision rules, and then evaluated their relative importance. This work is meaningful.

1 Authors should share the code and data of this work.

2 Figure 1 is so confused. Authors should update it.

3 It was suggested that MCC should be employed in this work.

4 The language should be polished by native English speakers.

5 Some efforts, such as 10.3389/fnins.2023.1197824, 10.1155/2022/9470683,can be discussed in this work.

6. PLOS authors have the option to publish the peer review history of their article (what does this mean?). If published, this will include your full peer review and any attached files.

Reviewer #1: No

---

## [Author Response · Author response to Decision Letter 0]

25 Sep 2023

Dear reviewer, 

Thank you for reviewing the manuscript Predicting Need for Heart Failure Advanced Therapies using an Interpretable Tropical Geometry-based Fuzzy Neural Network

 for publication in PLOS One. We sincerely appreciate the time and effort taken to review our paper and provide insightful and constructive comments. We carefully considered and addressed the comments and suggestions provided by the reviewers. We appreciate all the comments, as it helped us to redesign the experiments and improve the current proposed algorithm. Our point-by-point response to the reviewers’ comments and concerns is provided below, along with a tracked changed version of the manuscript that highlights all changes. We also produce a final version of the revised manuscript, with all line numbers included below referring to the final untracked version.

Response to Reviewer 1:

(1)

Comment:

Authors should share the code and data of this work.

Response

Thank you for pointing out the data and code availability. The code now is available on https://github.com/kayvanlabs/Generalized_fuzzy_neural_network_public. However, for the data, this study involves human research participant data and due to privacy and data sensitivity, it cannot be publicly shared. For researchers who meet the criteria for access to confidential data, data are available from the University of Michigan’s Innovation Partnerships (UMIP) office upon request (contact innovationpartnerships@umich.edu) 

(2)

Comment:

Figure 1 is so confused. Authors should update it.

Response

Thank you for your suggestions regarding the figure format. We have made updates to the figure to enhance its clarity in conveying the main framework and network structure. We changed the layout of the figure, add more details describing the experimental settings.

(3)

Comment:

It was suggested that MCC should be employed in this work.

Response

We appreciate your recommendation to include MCC as an additional evaluation metric. We have now computed MCC for all the models using both cross-validation and training-test schemes, and the results are included in the model performance tables.

(4)

Comment:

The language should be polished by native English speakers. 

Response

Thanks for your suggestions. The manuscript has been revised by Emily Wittrup, a senior computational biologist and native English speaker. Additionally, Dr. Jessica Golbus and Dr. Keith Aaronson, both native speakers, have reviewed and approved the revised manuscript.

(5)

Comment:

Some efforts, such as 10.3389/fnins.2023.1197824, 10.1155/2022/9470683, can be discussed in this work.

Response

Thanks for your suggestions. These two papers have been a great source of inspiration in the field of bioinformatics. We have incorporated a discussion of these two papers into the introduction section of the manuscript.

---

## [Decision Letter · Decision Letter 1]

14 Nov 2023

Predicting Need for Heart Failure Advanced Therapies using an Interpretable Tropical Geometry-based Fuzzy Neural Network

PONE-D-23-19605R1

Dear Dr. Zhang,

We’re pleased to inform you that your manuscript has been judged scientifically suitable for publication and will be formally accepted for publication once it meets all outstanding technical requirements.

Kind regards,

Suyan Tian

Academic Editor

PLOS ONE

Additional Editor Comments (optional):

Reviewers' comments:

Reviewer's Responses to Questions

**Comments to the Author**

1. If the authors have adequately addressed your comments raised in a previous round of review and you feel that this manuscript is now acceptable for publication, you may indicate that here to bypass the “Comments to the Author” section, enter your conflict of interest statement in the “Confidential to Editor” section, and submit your "Accept" recommendation.

Reviewer #1: All comments have been addressed

2. Is the manuscript technically sound, and do the data support the conclusions?

Reviewer #1: Yes

3. Has the statistical analysis been performed appropriately and rigorously? 

Reviewer #1: Yes

4. Have the authors made all data underlying the findings in their manuscript fully available?

Reviewer #1: Yes

5. Is the manuscript presented in an intelligible fashion and written in standard English?

Reviewer #1: Yes

6. Review Comments to the Author

Reviewer #1: Timely referral for advanced therapies (i.e., heart transplantation, left ventricular assist device) is critical for ensuring optimal outcomes for heart failure patients. Using electronic health records, our goal was to use data from a single hospitalization to develop an interpretable clinical decision-making system for predicting the need for advanced therapies at the subsequent hospitalization.

This work can be accepted.

7. PLOS authors have the option to publish the peer review history of their article (what does this mean?). If published, this will include your full peer review and any attached files.

Reviewer #1: No

---

## [Editor Report · Acceptance letter]

16 Nov 2023

PONE-D-23-19605R1 

Predicting Need for Heart Failure Advanced Therapies using an Interpretable Tropical
Geometry-based Fuzzy Neural Network 

Dear Dr. Zhang:

I'm pleased to inform you that your manuscript has been deemed suitable for publication in PLOS ONE. Congratulations! Your manuscript is now with our production department. 

Kind regards, 

on behalf of

Dr. Suyan Tian 

Academic Editor

PLOS ONE